# CagA toxin and risk of *Helicobacter pylori*-infected gastric phenotype: A meta-analysis of observational studies

**Cho Naing** [1]ᴏ*, **Htar Htar Aung** [2]ᴏ*, **Saint Nway Aye** [2], **Yong Poovorawan** [3], **Maxine A. Whittaker** [1]

1 College of Public Health, Medical and Veterinary Sciences, James Cook University, Queensland, Australia, 2 School of Medicine, IMU University, Kuala Lumpur, Malaysia, 3 Center of Excellence in Clinical Virology, Department of Pediatrics, Faculty of Medicine, Chulalongkorn University, Bangkok, Thailand

ᴏ These authors contributed equally to this work.
* cho3699@gmail.com (CN); htar.ng@gmail.com (HHA)

**Data Availability Statement:** All relevant data are within the manuscript and its Supporting information files.

## Abstract

### Background

*Helicobacter pylori* (*H. pylori*) is frequently associated with non-cardia type gastric cancer, and it is designated as a group I carcinogen. This study aimed to systematically review and meta-analyze the evidence on the prevalence of CagA status in people with gastric disorders in the Indo-Pacific region, and to examine the association of CagA positive in the risk of gastric disorders. This study focused on the Indo-Pacific region owing to the high disability adjusted life-years related to these disorders, the accessibility of efficient treatments for this common bacterial infection, and the varying standard of care for these disorders, particularly among the elderly population in the region.

### Methods

Relevant studies were identified in the health-related electronic databases including PubMed, Ovid, Medline, Ovid Embase, Index Medicus, and Google Scholar that were published in English between 1 January 2000, and 18 November 2023. For pooled prevalence, meta-analysis of proportional studies was done, after Freeman-Tukey double arcsine transformation of data. A random-effect model was used to compute the pooled odds ratio (OR) and 95% confidence interval (CI) to investigate the relationship between CagA positivity and gastric disorders.

### Results

Twenty-four studies from eight Indo-Pacific countries (Bhutan, India, Indonesia, Malaysia, Myanmar, Singapore, Thailand, Vietnam) were included. Overall pooled prevalence of CagA positivity in *H. pylori*-infected gastric disorders was 83% (95%CI = 73–91%). Following stratification, the pooled prevalence of CagA positivity was 78% (95%CI = 67–90%) in *H. pylori*-infected gastritis, 86% (95%CI = 73–96%) in peptic ulcer disease, and 83% (95%CI = 51–100%) in gastric cancer. Geographic locations encountered variations in CagA

**Funding:** The author(s) received no specific funding for this work.

**Competing interests:** The authors have declared that no competing interests exist.

prevalence. There was a greater risk of developing gastric cancer in those with CagA positivity compared with gastritis (OR = 2.53,95%CI = 1.15–5.55).

## Conclusion

Findings suggest that the distribution of CagA in *H. pylori*-infected gastric disorders varies among different type of gastric disorders in the study countries, and CagA may play a role in the development of gastric cancer. It is important to provide a high standard of care for the management of gastric diseases, particularly in a region where the prevalence of these disorders is high. Better strategies for effective treatment for high-risk groups are required for health programs to revisit this often-neglected infectious disease.

## Introduction

*Helicobacter pylori* (*H. pylori*) is a gram-negative bacterium that colonizes the stomach and is attributed to causing infections in humans [1, 2]. As *H. pylori* is frequently associated with non-cardia type of gastric cancer, and it is designated as a group I carcinogen [3]. Studies reported that only 20% of the colonizing bacteria at any time directly interact with the stomach epithelial cells [4], which are insufficient to cause gastric disorder [5]. This reflects that, in addition to its urease activity, *H. pylori* possesses additional virulence factors that are required to enable it to colonize and survive in the stomach's acid environment [4, 5]. The underlying mechanism of infectious diseases is greatly impacted by strain variations, which could account for the different outcomes of *H. pylori* infection. Studies have documented that *H. pylori* bacterial strains exhibit significant genetic diversity. Such diversity could result in an array of clinical outcomes, including gastritis, peptic ulcer disease, gastric adenocarcinoma (gastric cancer), and lymphoma. These outcomes depend on a variety of factors, particularly the virulence determinants expressed by distinct strains of *H. pylori* [6, 7]. Studies have reported that virulent determinants such as cytotoxin-associated gene A (CagA), lipopolysaccharide, vacuolating cytotoxin, peptidoglycan, and gamma-glutamyl transpeptidase are associated with gastric disease [8]. Among these, CagA is expressed by the terminal *cag*A gene of the type IV secretion system (T4SS) and discovered on the cag pathogenicity island (cagPAI) [9]. T4SS specifically forms a needle-like pilus device that is used to inject virulence factor such as the CagA effector protein into host target cells [10–13]. *H. pylori* strains with CagA-positive are more virulent than CagA-negative bacteria because they are linked to severe gastric inflammation [14].

With regards to gastric phenotype, there is an enigma coined as the "Asian Enigma" [15] or "Asian Paradoxical" [16] that affects many Asian countries [17, 18]. It draws attention to the need to study whether CagA prevalence is related to this paradoxical pattern. Several studies assessed CagA frequency in gastric disorders, which showed different findings. Since *H. pylori*-infected patients showed variations in the clinical manifestations such as gastritis, peptic ulcers, and gastric cancer, reporting the CagA prevalence on a range of gastric disorders is of immense value. Additionally, a recent review of the Global Burden of Disease data reported that Indo-Pacific super-, region has the highest disability-adjusted life years (DALY) burden for more digestive disease types, including oesphageal disorders and ulcers (17%) globally. These disorders can cause considerable distress and disability for those who suffer from them, as well as a large burden on the health care system and its financing [19]. Moreover, some socioeconomic groups within the countries of this region have been shown to have poorer 'Quality of Care Index' (e.g. access to effective treatment) for peptic ulcer disease [20]. It is

imperative to emphasize the significance of a common and treatable bacterial infectious disease [21] as a cause of peptic ulcer disease in these settings, as countries struggle with the rising expenses of providing universal health care for their citizens and an aging population that is frequently underserved for gastric disorders, including peptic ulcers [20].

Taking all these into consideration, the objectives of this study were to systematically review and meta-analyze the evidence on the prevalence of CagA status in people with gastric disorders in the Indo-Pacific region, and to examine the association of CagA positive in the risk of gastric disorders.

## Materials and methods

This study followed the PRISMA-2020 checklist [22] [S1 Checklist]. The protocol of this study is available from the corresponding author upon reasonable request. The consent from the participants was not necessary as the current study used solely published data.

### Search strategy

Two investigators (CN, HHA) looked at the relevant studies in health-related electronic databases including PubMed, Ovid, Medline, Ovid Embase, Index Medicus, and Google Scholar. The search terms used were (*Helicobacter pylori*, *H. pylori*, virulence factor, CagA, gastric cancer, gastritis, and peptic ulcer) with appropriate Boolean operators (AND/OR). The search was confined to publications in English between January 1 2000 and November 18 2023. Furthermore, hand-searching took place following the same inclusion criteria set for this study. Search Strategy in PubMed is provided in S1 Table.

### Selection criteria

Studies were considered if they met the following criteria.

1. Human studies of observational design that assessed adults with *H. pylori*-infected gastric disorder.

2. Conducted in low- and middle-income Indo-Pacific countries.

3. Assessed the CagA status, irrespective of the detection method.

4. Provide sufficient data to compute the prevalence of CagA, and/or association of CagA with any or all three gastric disorders (i.e., gastritis, peptic ulcer disease, gastric cancer)

5. Examined the presence of genotypes with the use of PCR.

6. Included a minimum of ten samples.

In this study, the participants with duodenal and/or gastric ulcerations were categorized as peptic ulcer disease as stated in the primary studies. Endoscopic gastritis was regarded as an equivalent to 'non-ulcer disorder' with neither a history of ulcer disease nor endoscopic evidence of ulcer disease [23].

### Exclusion criteria

Studies that did not satisfy the criteria for inclusion were not considered. In addition, reviews, letters, editorials, efficacy trials, and modelling studies were omitted.

## Data collection

One investigator (CN) screened the title and abstracts, which were deemed eligible. Two investigators (HHA, CN) independently selected studies from those saved from the databases. Full-text studies that were deemed appropriate were further reviewed. Any discrepancy between the two investigators was settled by agreement. Using a self-created data extraction sheet, one investigator (CN) gathered data from the eligible studies. Another investigator (HHA) cross-checked this. Data collection included first author, year of publication, study country, study design, prevalence (numerator and denominator), participant characteristics (age, sex), types of gastric disorders (gastritis, peptic ulcer, gastric cancer), and methods of DNA sequencing.

## Assessment of the quality of studies

The methodological quality of the studies involved in this study was rated using the Newcastle-Ottawa scale (NOS) checklist [24]. This tool has a star rating as a ranking scheme for observational studies, where nine stars are the maximum obtainable. The higher star ranking obtained, the better quality of the study.

## Data analysis

At the individual-study level, the prevalence (along with variance) of CagA in a certain gastric phenotype was estimated by dividing the number of CagA-positive cases by the total number of cases examined. For pooling of studies, a proportional meta-analysis was carried out as described by Barker and associates [25], after variance normalization with the Freeman-Tukey double arcsine transformation [26]. For the association of CagA and the development of gastric disorder, a pooled odds ratio (OR) with 95% CI was estimated.

Concerning the variations among studies, the DerSimonian and Laird random effect model was employed for computing summary estimates [27]. An $I^2$ value of >75% denotes substantial variation [28]. Publication bias was not assessed for proportional meta-analysis owing to a methodological shortcoming in Egger's tests and funnel plots [25]. As there was a sufficient number of studies, Egger's test was used to gauge publication bias for the risk comparisons between gastritis and peptic ulcers. A simple test of asymmetry of the funnel plot (plot of effect estimates against sample size) was visualized [29].

Data analyses were done with the "*metaprop_one*" command for proportional meta-analysis, "*metan*" command for binary associations, and "*meta*" for publication bias command in STATA (version 15) (TXT, USA).

## Results

Fig 1 illustrates the PRISMA flow diagram indicating the study selection process in the present meta-analysis. A total of 24 studies (26 datasets) were identified across eight Indo-Pacific countries such as Bhutan, India, Indonesia, Malaysia, Myanmar, Thailand, Singapore, and Vietnam [30–53]. Publication years spanned from 2005 [46] to 2023 [42]. A list of excluded studies along with their justifications is provided in S2 Table.

Table 1 presents the main characteristics of the 24 studies included in the current meta-analysis. Most of the studies were conducted in Thailand (6 studies) and Malaysia (6 studies). These individual studies examined a variety of gastric disorders, the most common being gastritis (23 studies), followed by peptic ulcer disease (17 studies), and gastric cancer (11 studies). The Trang study reported data from three countries, Bhutan, Myanmar and Vietnam [49]. Fifteen studies reported data on comparison of CagA positive and negatives in gastric disorders [31, 33–37, 41–46, 49, 52, 53]. The methodological quality of the studies identified was a

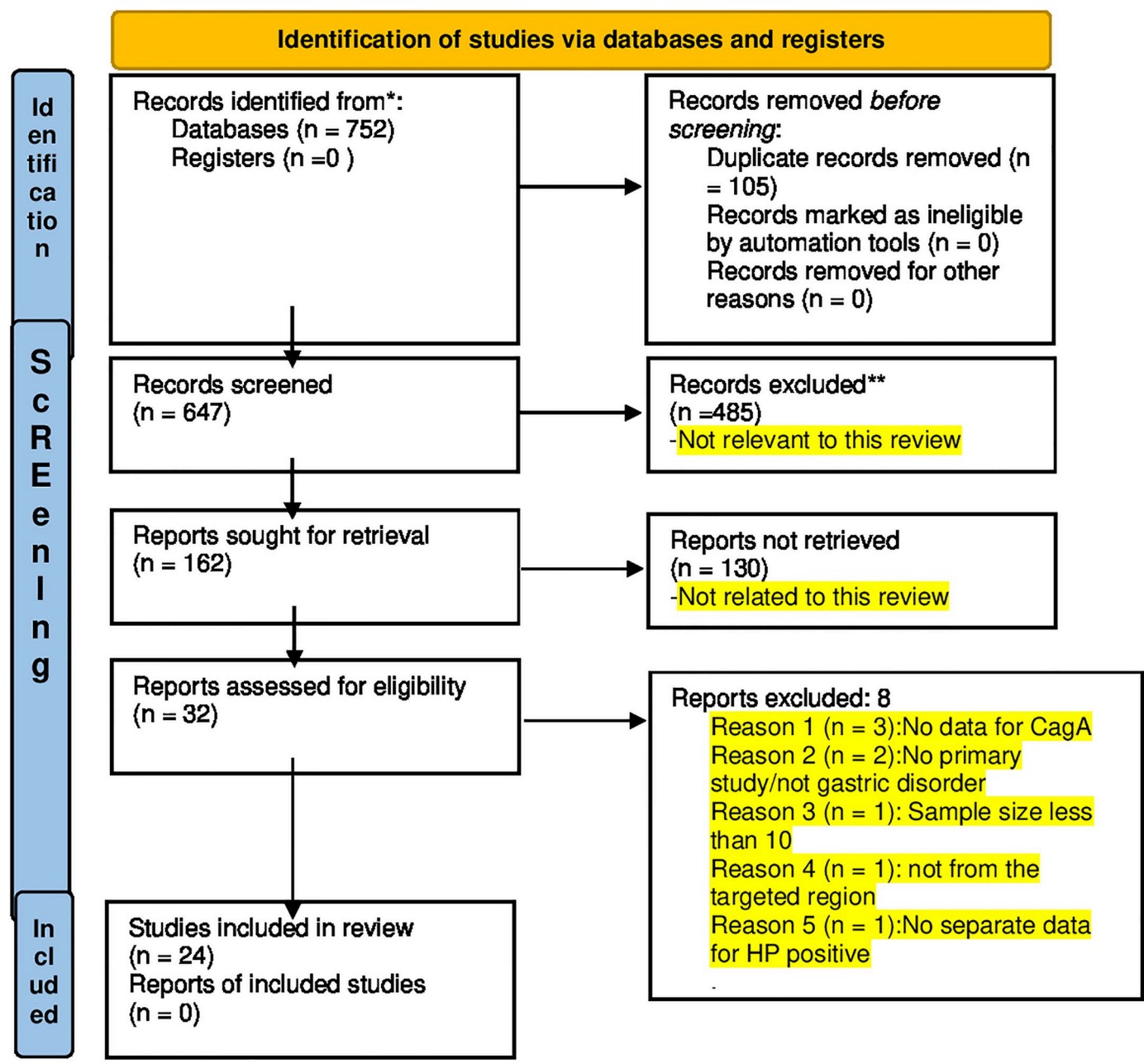

**Fig 1. PRISMA flowchart indicating the study selection process.**

median of 6 stars (range 5–8 stars), reflecting most of the studies were of moderate quality (S3 Table).

## Prevalence of CagA in *H. pylori*-infected gastric disorders

The CagA prevalence in *H. pylori*-infected gastric disorders was assessed in 24 studies, irrespective of the specific type (Fig 2). The overall pooled prevalence of CagA in *H. pylori*-infected gastric disorders was 83% (95%CI = 73–91%). This indicated that the great majority of

**Table 1. Characteristics of the studies identified (N = 24).**

| No. | Study [ref #] | Yr | Country | Design | Mean age, yr | Male n(%) | Total | CagA+ve/tested samples | | | Remark |
|-----|---------------|----|---------|--------|--------------|-----------|-------|------|-----|-----|--------|
| | | | | | | | | GT | PUD | GC | |
| 1 | Myint [40] | 2018 | Myanmar | CS | 40.1±11. | 28 (57) | 69 | 61/69 | NA | NA | 3DU+1 GC |
| 2 | Ansari [32] | 2017 | Myanmar | CS | 40.7± 11.5 | 30 (24) | 72 | 67/72 | NA | NA | GT 89.3%;a subset data |
| 3 | Jeyamani [36] | 2018 | India | CS | NA | NA | 61 | 16/35 | 12/14 | 0/0 | |
| 4 | Linpisarn [37] | 2007 | Thailand | CS | NA | NA | 135 | 48/58 | 68/73 | 3/4 | |
| 5 | Boonyanugomol [33] | 2020 | Thailand | CS | NA | NA | 80 | 38/50 | 17/20 | 8/10 | |
| 6 | Nguyen [42] | 2023 | Vietnam | CS | 9.1± 2.4 | NA | 269 | 0 | 185 | NA | |
| 7 | Nguyen [41] | 2010 | Vietnam | CS | NA | NA | 100 | 71 | 24 | NA | |
| 8 | Pandya [43] | 2017 | India | CS | R:10–90 | NA | 43 | 9/34 | 6/9 | NA | |
| 9 | Chomvarin [34] | 2008 | Thailand | CS | 49.5 | 55 (49) | 112 | 60/62 | 34/34 | 16/16 | |
| 10 | Chomvarin [35] | 2012 | Thailand | CS | 50 (R: 15–81 | 77 (52) | 147 | 65/68 | 52/57 | 17/18 | Mixed/ unclassified not included. |
| 11 | Yamada [52] | 2013 | Thailand | CS | 56 ± 11.3[1] | 93(42) | 220 | 34/134 | NA | 7/86 | Benign = GT; 1 = GC group |
| 12 | Zheng [53] | 2000 | Singapore | CS | NA | NA | 108 | 37/41 | 58/67 | NA | |
| 13 | Trang [49] | 2015 | Vietnam | CS | 44.1 ± 12.7 | 47 (46) | 102 | 72/76 | 25/26 | NA | |
| | | | Myanmar | CS | 40.1 ± 11.7 | 28 (41) | 66 | 59/66 | NA | NA | |
| | | | Bhutan | CS | 36.8 ± 13.4 | 94 (47) | 200 | 161/161 | 38/38 | NA | |
| 14 | Uchida [51] | 2009 | Thailand | CS | 45 | 47 (45) | 103 | 98 | NA | NA | 25 = PUD |
| 15 | Alfizah [30] | 2012 | Malaysia | CS | 56.05 ± 12.95 | 58 (53) | 110 | 81/81 | 28/28 | NA | |
| 16 | Mohamed [39] | 2009 | Malaysia | CS | NA | NA | 93 | 44/93 | 26/93 | NA | Based on the total tested. |
| 17 | Miftahussurur [38] | 2015 | Indonesia | CS | 47.8 ± 14.6 | NA | 44 | 43 | 0 | 0 | 8 Other types are included. |
| 18 | Tiwari [48] | 2011 | India | CS | NA | NA | 32 | 3 | 15 | 14 | |
| 19 | Ali [31] | 2005 | India | CS | 48.4 | 100 (57) | 174 | 4/56 | 53/104 | 12/14 | |
| 20 | Tan [46] | 2005 | Malaysia | CS | 55.53 ± 12.52 | 72(57) | 127 | 59/75 | 25/28 | 0/2 | |
| 21 | Tan [47] | 2006 | Malaysia | CS | 53.38 ± 12.09 | 36 (49) | 73 | 58/73 | NA | NA | |
| 22 | Truong [50] | 2009 | Vietnam | CS | NA | NA | 22 | NA | 11/11 | 11/11 | |
| 23 | Schmidt [44] | 2009 | Malaysia | CS | NA | NA | 127 | 105/105 | NA | 21/21 | |
| 24 | Schmidt [45] | 2010 | Malaysia | CS | NA | 87 (55) | 159 | 75/75 | 16/16 | 22/22 | |

DU: duodenal ulcer; GC: gastric cancer; GT: gastritis; PUD: peptic ulcer disease; NA: not available; yr: year

population in Indo-Pacific region who had *H. pylori*-infected gastric disorders were CagA positive. On stratification, Vietnam (90%, 95%CI = 73–99%; 4 studies) had the highest pooled prevalence of CagA, followed by Malaysia (81%, 95%CI = 68–91%; 6 studies), Thailand (81%, 95%CI = 48–99%; 6 studies), Myanmar (90%, 95%CI = 85–93%; 3 studies), and India (60%, 95%CI = 34–84%; 4 studies). Notably, all except the studies in Myanmar ($I^2$: 0%). showed substantial heterogeneity ($I^2$: > 93%). There was only a single study from Bhutan (100%, 95% CI = 98–100%), Indonesia (98%, 95%CI = 88–100%) and Singapore (88%, 95%CI = 80–93%).

## CagA prevalence in *H. pylori*-infected gastritis

On stratification by type of gastric disorders, CagA prevalence in *H. pylori*-infected gastritis was reported in 23 studies. Overall, the summary prevalence of CagA in gastritis was 79% (95%CI = 67–90%). Vietnam (94%, 95%CI = 90–97%; 2 studies) had the highest pooled prevalence, followed by Malaysia (88%, 95%CI = 69–99%; 6 studies), Thailand (82%, 95%CI = 56–98%; 6 studies), Myanmar (88%, 95%CI = 82–93%; 2 studies), and India (20%, 95%CI = 6–40%; 4 studies). Heterogeneity was substantial among studies in Malaysia and Thailand ($I^2$: > 96%), but it was absent among studies in India, Myanmar, and Vietnam. Only a single study was

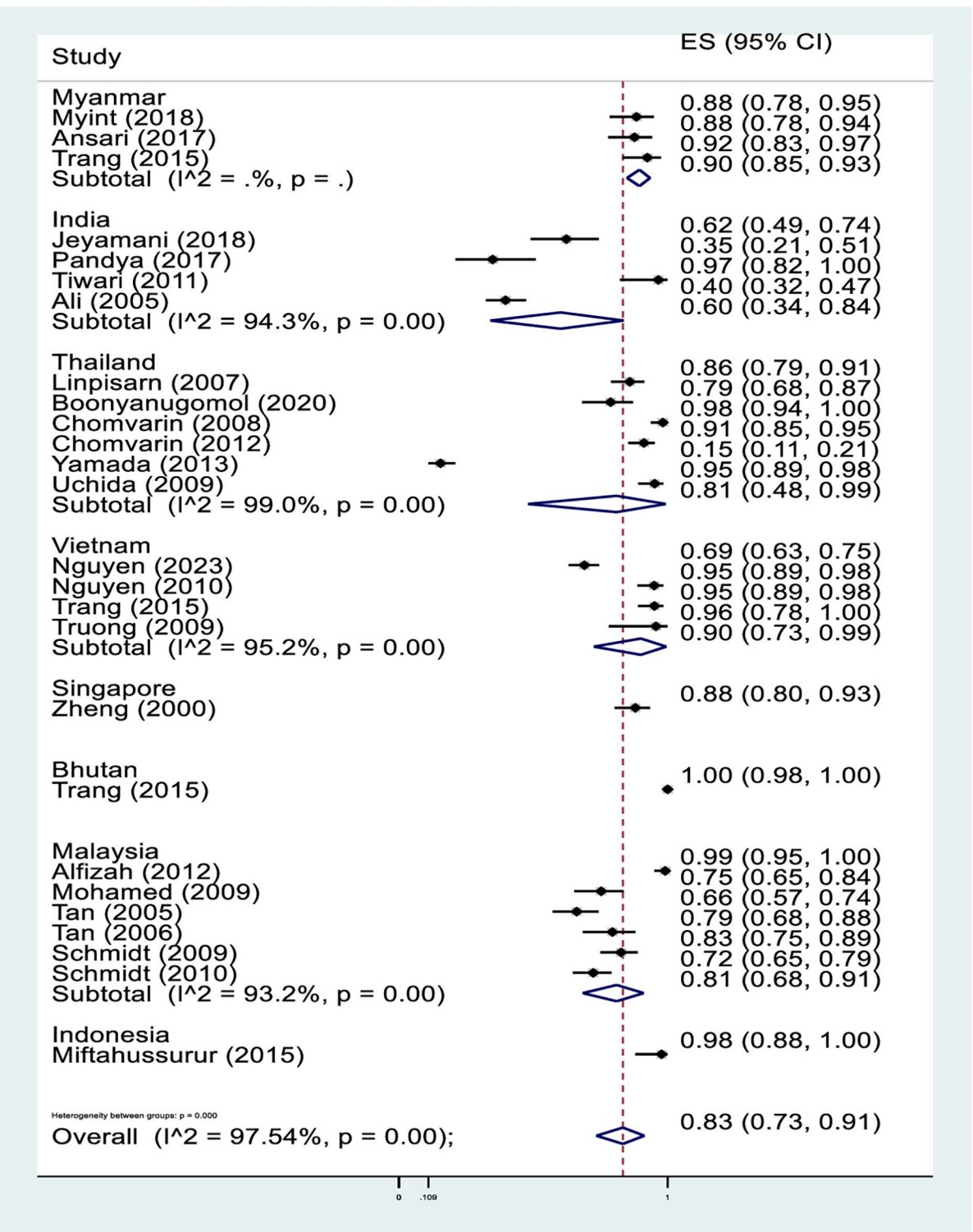

**Fig 2. Pooled prevalence of CagA in *H. pylori*-infected gastric disorders.** Note: A horizontal line represents individual study. A point estimate (prevalence) of an individual study is represented by the square in the line. The line's width indicates 95% confidence interval. A pooled prevalence is represented by the diamond. Proportion is the estimated percentage of prevalence. The $I^2$ value in percentage denotes the degree of heterogeneity between studies. For example, a pooled prevalence of CagA in Myanmar is 90% (95%CI:85–93%, $I^2$:0%).

available either from Bhutan (100%, 95%CI = 98–100%), Indonesia (98%, 95%CI = 88–100%) or Singapore (90%,95%CI = 77–97%) (S1 Fig).

## CagA prevalence in *H. pylori*-infected peptic ulcer disease

Seventeen studies reported data on CagA prevalence among participants with *H. pylori*-infected peptic ulcer disease. Overall, the summary prevalence of CagA in participants with *H. pylori*-infected peptic ulcer disease was 86% (95%CI = 73–96%). The highest summary prevalence of CagA was reported from Thailand 94% (95%CI = 87–99%; 4 studies), followed by Vietnam (91%, 95%CI:50–100%; 4 studies) and Malaysia (86%, 95%CI = 37–100%, 4 studies). India accounted for a prevalence of 59% (95%CI = 43–75%; 4 studies). Heterogeneity was substantial among these studies in India, Malaysia, Thailand, and Vietnam ($I^2$: >57%). The single studies from Bhutan (100% (95%CI = 91–100%) and Singapore (87%, 95%CI = 76–94%) reported a prevalence of CagA in participants with peptic ulcer disease (S2 Fig).

## CagA prevalence in *H. pylori*-infected gastric cancer

A total of 11 studies reported data on CagA prevalence among participants with *H. pylori*-infected gastric cancer. Overall, 83% (95%CI = 51–100%, $I^2$: 95%) of patients with gastric cancer had CagA positives. On stratification by countries, Malaysia had the highest prevalence of CagA (96%, 95%CI = 87–100%, $I^2$: 0%; 2 studies), followed by India (90%, 95%CI = 75–99%; $I^2$: 0%; 2 studies), and Thailand (71%, 95%CI = 16–100%%; $I^2$: 96%; 5 studies). Notably, heterogeneity was substantial among studies in Thailand ($I^2 > 98$%), while studies in India and Malaysia had the absence of heterogeneity. A single study was available from Vietnam (92%, 95%CI = 65–99%) (S3 Fig).

## Associations between CagA toxin and the risk of gastric disorders

Table 2 provides data on the association studies. Fifteen studies were available for an assessment of the relationship between CagA and gastric disorder [31, 33–37, 41–46, 49, 52, 53]. Of

**Table 2. Distribution of CagA in gastric phenotype.**

| Sr. no. | Study | Year | Country | GT+ (total) | PUD+(total) | GC+(total) |
|---|---|---|---|---|---|---|
| 1 | Ali [31] | 2005 | India | 4 (56) | 49 (104) | 12(14) |
| 2 | Boonyanugomol [33] | 2020 | Thailand | 38 (50) | 17 (20) | 8 (18) |
| 3 | Chomvarin [34] | 2008 | Thailand | 60 (62) | 34 (68) | 16 (32) |
| 4 | Chomvarin [35] | 2012 | Thailand | 65 (68) | 52 (109) | 17 (35) |
| 5 | Jeyamani [36] | 2018 | India | 21 (42) | 12 (14) | NA |
| 6 | Linpisarn [37] | 2007 | Thailand | 48 (58) | 68 (73) | 3 (7) |
| 7 | Nguyen [41] | 2010 | Vietnam | 71 (76) | 24 (24) | NA |
| 8 | Nguyen [42] | 2023 | Vietnam | NA | 185 (268) | NA |
| 9 | Pandya [43] | 2017 | India | 9 (34) | 3 (12) | NA |
| 10 | Schmidt [44] | 2009 | Malaysia | 103 (105) | NA | 21 (42) |
| 11 | Schmidt [45] | 2010 | Malaysia | 121 (121) | 16 (48) | 22 (42) |
| 12 | Tan [46] | 2005 | Malaysia | 59 (75) | 25 (53) | NA |
| 13 | Trang [49] | 2015 | Vietnam | 72 (76) | 25 (51) | NA |
| | | | Bhutan | 161(161) | 38 (38) | NA |
| 14 | Yamada [52] | 2013 | Thailand | 71 (76) | 24 (48) | NA |
| 15 | Zheng [53] | 2000 | Singapore | 58 (77) | 37 (78) | NA |

GC: gastric cancer; GT: gastritis; NA: not available/not reported; PUD: peptic ulcer disease.

these, 13, seven, and six studies compared the risk between peptic ulcer disease and gastritis, gastric cancer and gastritis, and peptic ulcer disease and gastric cancer, respectively.

Thirteen studies were available for a comparison of the CagA positive between peptic ulcer disease and gastritis risks [31–37, 41, 43–46, 49, 52, 53]. Only six studies were available for a comparison of the CagA positive between gastric cancer and gastritis [33–35, 37, 44, 45]. Five studies reported data for all three types of gastric disorder [33–35, 37, 44].

Overall, CagA was a comparable risk for both peptic ulcer disease and gastritis (OR 1.03, 95%CI = 0.37 to 2.91, 1763 participants, 13 studies, $I^2$: 82%). There were no differences in the risks in any subgroup of countries (Fig 3). Overall, presence of CagA was associated with an increased risk of developing gastric cancer than gastritis (OR 2.53, 95%CI = 1.15 to 5.53, 613 participants, 7 studies, $I^2$: 78%) (Fig 4), but a comparable risk for both gastric cancer and peptic ulcer disease (OR 0.44, 95%CI = 0.08 to 2.34, 404 participants, 6 studies, $I^2$: 67%) (Fig 5). Notably, in all these comparisons, there were substantial heterogeneities among the studies.

## Publication bias

A funnel plot showed asymmetry, indicating the presence of publication bias in comparison between peptic ulcer disease and gastric cancer (13 studies included) (Fig 6). For other comparisons (peptic ulcer diseases vs gastritis; gastric cancer vs gastritis) it was not possible to perform Eggers's test as they had less than the ten studies recommended for this assessment.

## Discussion

This study presents information on the pooled prevalence of CagA among those with *H. pylori*-infected gastric disorders based on the data from 24 observational studies across eight countries in the Indo-Pacific region. According to the findings, distinct *H. pylori* CagA strains are circulating with geographical variation. The global prevalence of CagA-positive *H. pylori* ranges from 43% to 90% [54]. and that was comparable with the current findings of 82% in gastric disorders. After recalculation of data provided in a published review including four Asian countries [55], CagA prevalence of 84% (95%CI = 67–100%) was also comparable with the current findings, which encompassed 24 studies from eight Indo-Pacific countries (82%, 95%CI = 72–99%).

Stomach mucosa in the antrum and corpus almost always becomes infiltrated by neutrophilic and mononuclear cells during *H. pylori* colonization [56]. The primary condition associated with *H. pylori* colonization is chronic active gastritis, and additional conditions specifically linked to *H. pylori* are the result of this continuing inflammatory process [5, 56]. Hence, our analysis of the risk of developing peptic ulcer disease or gastric cancer compared to gastritis is meaningful. Compared with gastritis, the presence of CagA had an increased risk of developing gastric cancer observed in this study could be explained with biological plausibility.

The difference in the occurrences of CagA of gastric cancer in India, Thailand, and Malaysia may be explained by the phenomena of various strains circulating within different ethnic groups, in conjunction with host and specific environmental factors. As described elsewhere [57], *H. pylori* may have arrived in Asia as part of the five ancient human migrations. The history of human population expansion in this region has been influenced by multiple migrations and population expansions.

Molecular quantitative analysis showed that the load of CagA injected after one hour was approximately 16% of the total CagA existing in the adjacent bacteria, and 30% after three hours [58]. Considering of the low percentage of the total amount of bacteria in the infection and the load of CagA injected by the bound bacteria, this might be one of many reasons why

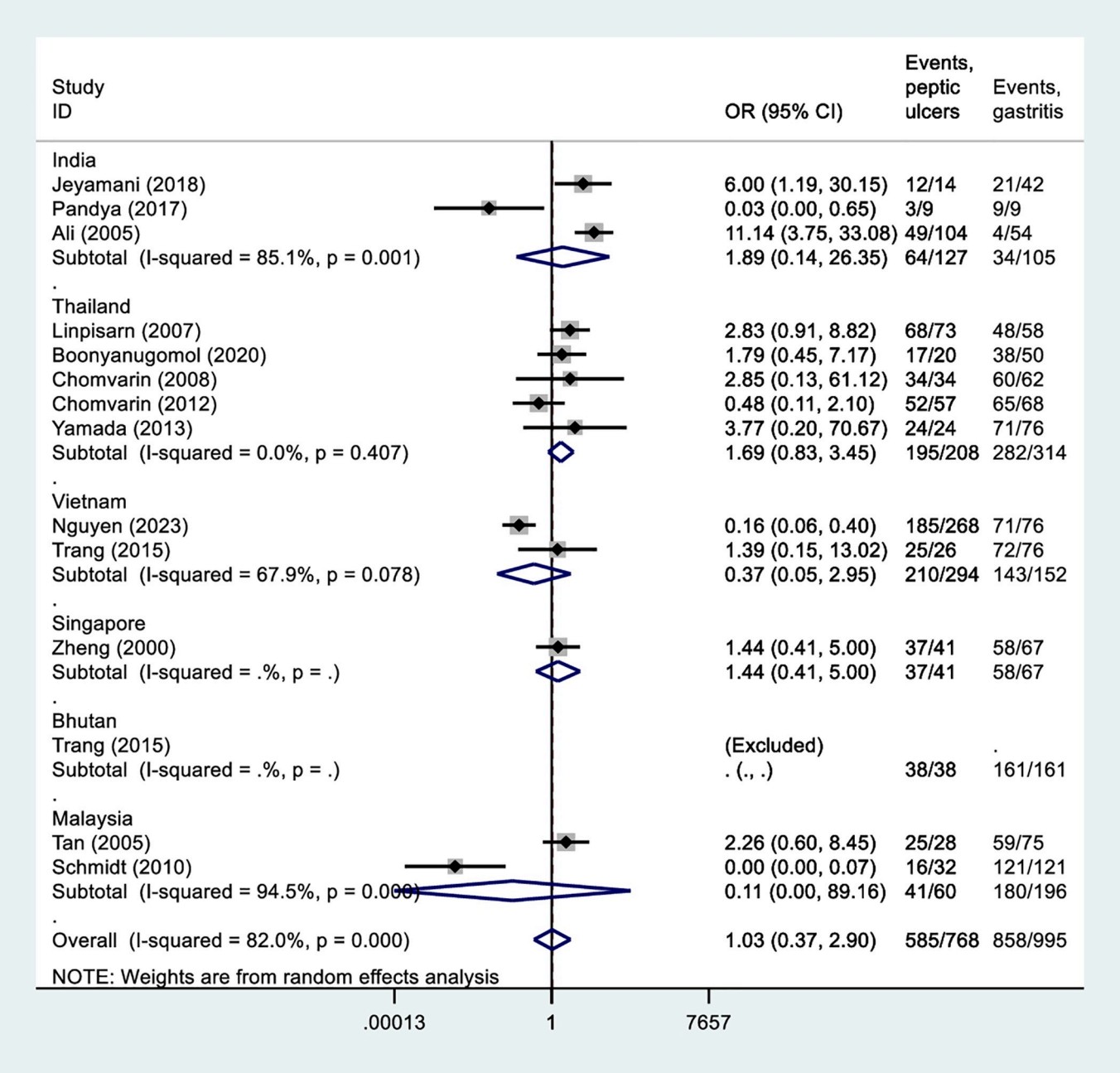

**Fig 3. Forest plot indicating the association of CagA in peptic ulcer disease and gastritis.**

there was low occurrence of gastric cancer in the countries with high incidence of *H. pylori* infection.

CagA in the development of gastric disorder is not yet fully understood. There are many ways in which translocated CagA interferes with host cell functions. For instance, the CagA–PAR1 interaction [59], the interaction between SHP2 and gp130 (Src homology 2 domain-containing protein tyrosine phosphatase 2 and glycoprotein 130) [60], the interaction of the IL6/gp130/STAT3 (interleukin-6/glycoprotein 130/signal transducer and activation of

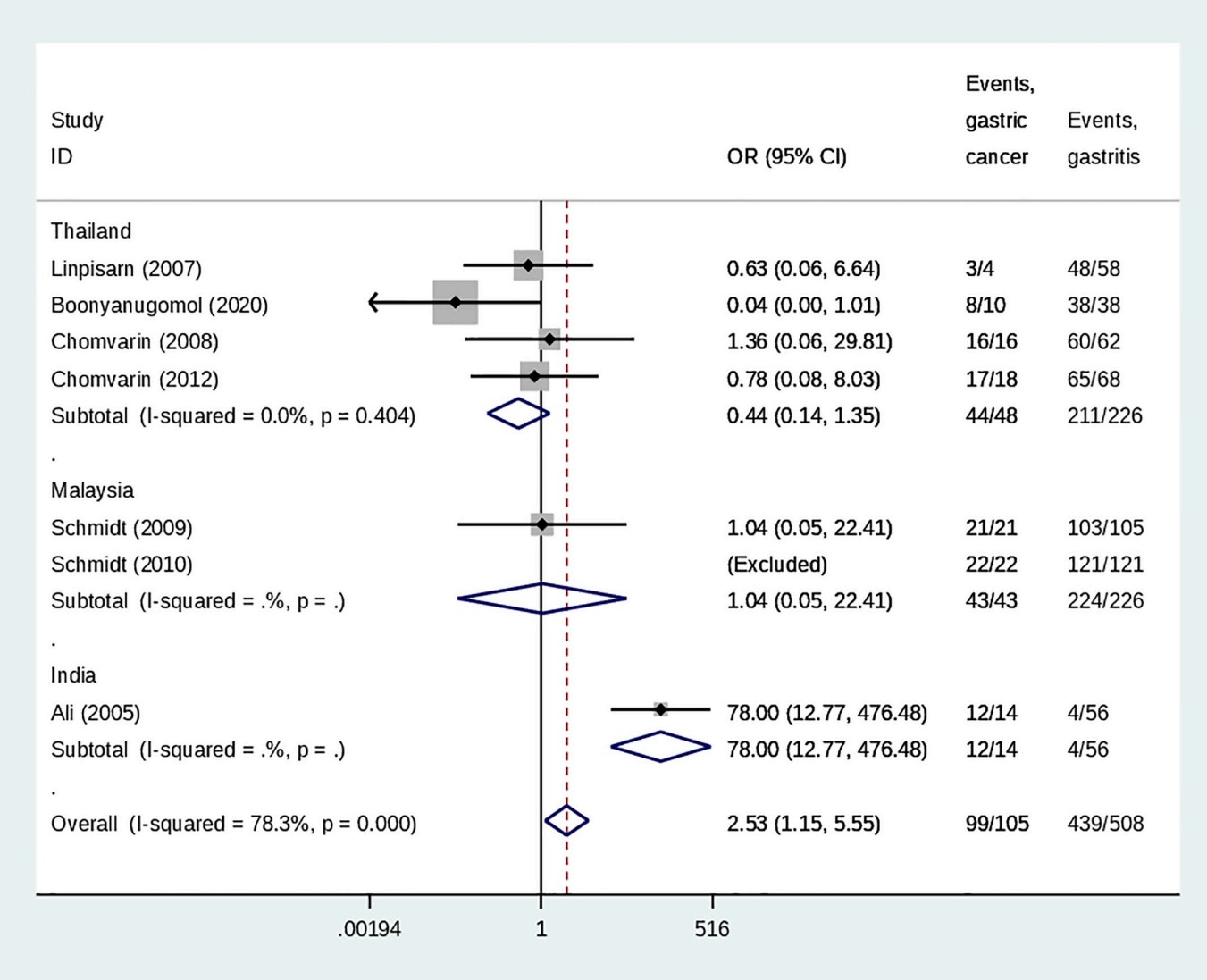

**Fig 4. Forest plot indicating the association of CagA in gastric cancer and peptic ulcer disease.**

transcription 3) pathway [60, 61], inflammation attribution to the changes in chemokine profile of tissue [62], among others. Like previous meta-analyses [63, 64], the current study documented that CagA toxin of *H. pylori* may contribute to gastric cancer.

## Study limitation

There are some limitations. This study only reviewed published studies written in English. As such, there is an issue of information bias because it is likely to miss unpublished studies or the non-English publications that may have been appropriate to include. Regarding an inherent limitation of cross-sectional/survey studies, the prevalence estimates in the primary studies identified for the present analysis could vary as times passes. Considering the associations observed in the current meta-analysis, it would be crucial to substantiate the results using cohort studies with extended follow-up times. The merit of observational studies is that they

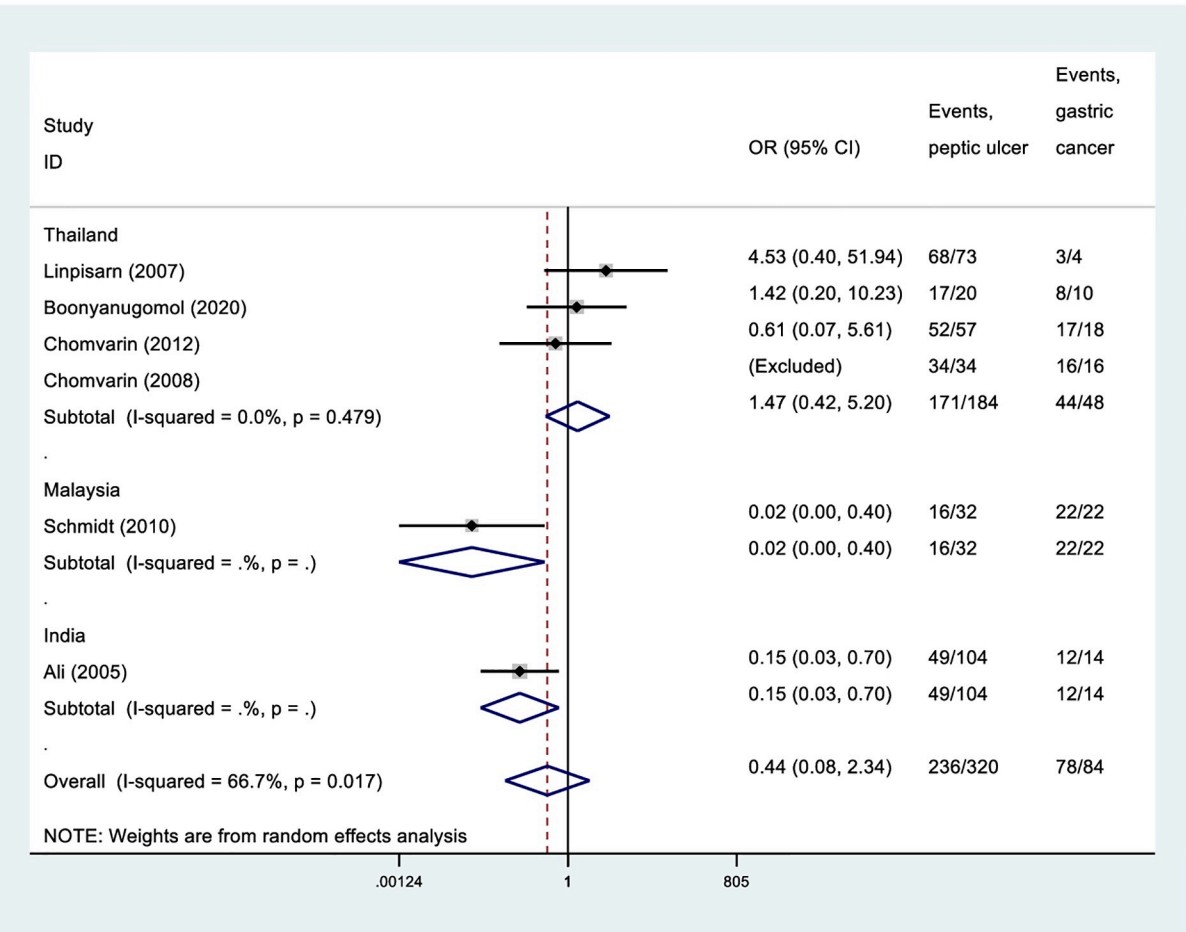

**Fig 5. Forest plot indicating the association of CagA with gastric cancer and peptic ulcer disease.**

can determine health problems existing, reflecting real-life conditions. Due to the paucity of data, subgroups such as the age of the study population, and unique types of gastric cancer (e.g., intestinal, cardina, diffuse) could not be analyzed. The present study included single genotype CagA. Hence, selection bias is a concern. Future studies should consider multiple genotypes in analysis. Due to publication bias related to small studies that tended to exaggerate the true effect [65], as well as limited number of studies that could compare CagA between the risks of gastric cancer and peptic ulcer disease, a type II error may be an issue. The estimated prevalence in this study with substantial heterogeneity might not be related to sampling error only [66]. High heterogeneity may be observed albeit with more homogenous subgroups such as countries. Therefore, a high $I^2$ value is not a signal that the data are inconsistent [25]. Thus, these cautions should be considered while interpreting the findings.

## Implications

The higher incidence of CagA in peptic ulcer disease suggests any *H. pylori* infection prevention program can help preventing gastritis from progressing to peptic ulcers, and gastric cancer a more costly and potentially fatal result of a common and treatable bacterial infection. Additionally, a higher risk of stomach cancer affects the capacity to access efficient treatments

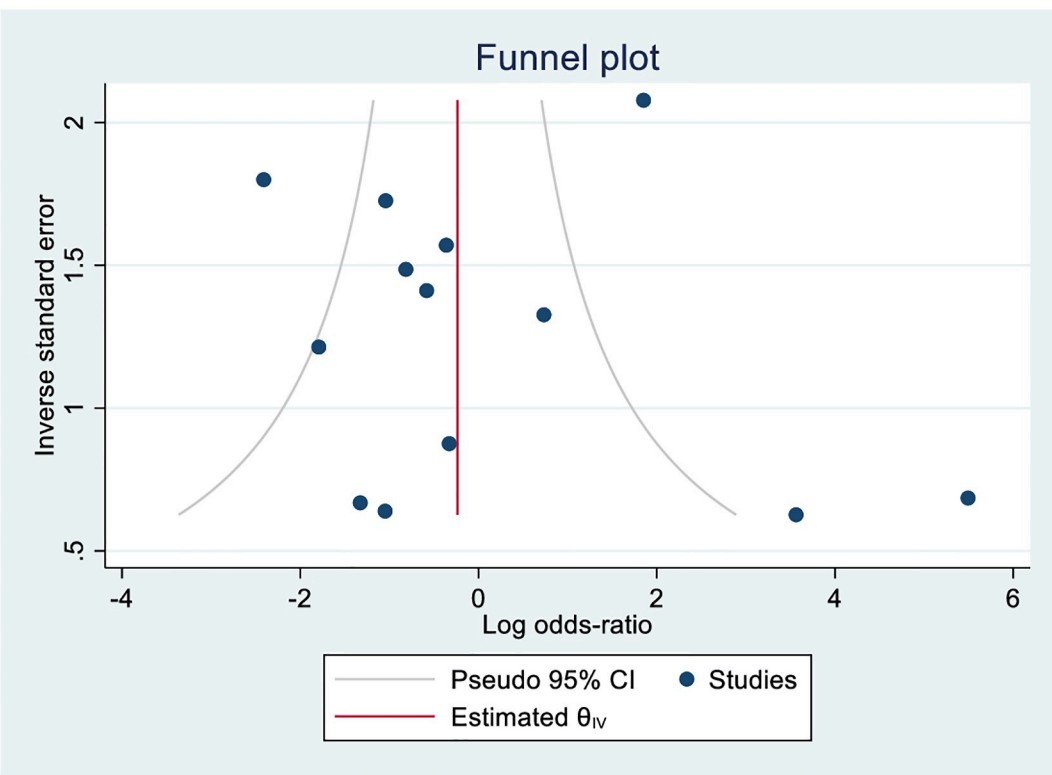

**Fig 6. Funnel plot asymmetry showing publication bias.**

that could lower the risk of stomach cancer. Given that different countries have distinct risks that reflect differences in their ethnic composition, identifying and prioritizing the high-risk population will help with better planning and prioritization of health programs to manage this cluster of disorders associated with *H. pylori* and may also make services more accessible and affordable. Reiterating the substantial evidence of association between a treatable infection and a high burden of disease and health care utilisation can help improve equity in their programs during this second half of the Sustainable Development Goal period, when countries in the region are struggling to attain the universal health coverage goal set [20].

## Conclusion

Findings suggest that the distribution of CagA in *H. pylori*-infected gastric disorders varies among gastric disorders in the study countries, and CagA has a plausible role in the development of gastric cancer. It is important to the quality of care for the management of gastric diseases, particularly in a region where the prevalence of these disorders is high. Better strategies for effective treatment for high-risk groups are required for health programs to revisit this often-neglected infectious disease.

## Supporting information

**S1 Checklist. PRISMA-2020 checklist.**
(DOC)

**S1 Table. Search strategy.**
(DOC)

**S2 Table. Excluded studies and the reasons for exclusion.**
(DOC)

**S3 Table. Quality assessment with the NOS assessment criteria.**
(DOC)

**S1 Fig. Pooled prevalence of CagA in *H. pylori*- infected gastritis.**
(DOC)

**S2 Fig. Pooled prevalence of CagA in *H. pylori*- infected peptic ulcer disease.**
(DOC)

**S3 Fig. Pooled prevalence of CagA in *H. pylori*- infected gastric cancer.**
(DOC)

# Acknowledgments

We gratefully acknowledge the participants and authors of the primary studies that are included in this study. We thank the editors and anonymous reviewers for the comments and helpful input, and our institutions for allowing us to perform this study.

# Author Contributions

**Conceptualization:** Yong Poovorawan, Maxine A. Whittaker.

**Data curation:** Cho Naing, Htar Htar Aung, Saint Nway Aye.

**Formal analysis:** Cho Naing, Htar Htar Aung, Saint Nway Aye.

**Methodology:** Cho Naing, Htar Htar Aung, Yong Poovorawan, Maxine A. Whittaker.

**Supervision:** Maxine A. Whittaker.

**Writing – original draft:** Cho Naing, Htar Htar Aung.

**Writing – review & editing:** Cho Naing, Maxine A. Whittaker.

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
