## [Decision Letter · Decision Letter 0]

4 Jun 2024

PONE-D-24-13758CagA toxin and risk of Helicobacter pylori-infected gastric phenotype: A meta-analysis of observational studiesPLOS ONE

Dear Dr. Naing,

Thank you for submitting your manuscript to PLOS ONE. After careful consideration, we feel that it has merit but does not fully meet PLOS ONE’s publication criteria as it currently stands. Therefore, we invite you to submit a revised version of the manuscript that addresses the points raised during the review process.

We look forward to receiving your revised manuscript.

Kind regards,

Tomasz M. Karpiński, Prof.

Academic Editor

PLOS ONE

Reviewers' comments:

Reviewer's Responses to Questions

**Comments to the Author**

1. Is the manuscript technically sound, and do the data support the conclusions?

Reviewer #1: Yes

Reviewer #2: Yes

Reviewer #3: Partly

2. Has the statistical analysis been performed appropriately and rigorously? 

Reviewer #1: Yes

Reviewer #2: Yes

Reviewer #3: No

3. Have the authors made all data underlying the findings in their manuscript fully available?

Reviewer #1: Yes

Reviewer #2: Yes

Reviewer #3: Yes

4. Is the manuscript presented in an intelligible fashion and written in standard English?

Reviewer #1: Yes

Reviewer #2: No

Reviewer #3: No

5. Review Comments to the Author

Reviewer #1: Dear Editorial office

It is my honour to read and review this nicely entitled paper by salomon et al “CagA toxin and risk of Helicobacter pylori-infected gastric phenotype: A meta-analysis of observational studies” submitted to Plos one. In principle I found it suitable for publication here pending your corrections.

1- The title seems fine and no change warranted.

2- Abstract is too general-like, so please enrich it

3- Abstract “To confirm these findings, well-designed studies with sufficient samples are recommended.” Is not absolutely a good text for here, remove it.

4- Page 4, line 76, why there are two major types of reference methods? “epithelial cells [4],4 which” revise it for revision.

5- Introduction section sentence “This reflects that, in

addition to its urease activity, H. pylori possesses additional virulence factors that are required to enable it to colonize and survive in the stomach's acid environment.” We need the below text references

** Abadi, A. T. B. (2017). Strategies used by helicobacter pylori to establish persistent infection. World journal of Gastroenterology, 23(16), 2870.

*** Israel, Dawn A., and Richard M. Peek Jr. "The role of persistence in Helicobacter pylori pathogenesis." Current opinion in gastroenterology 22.1 (2006): 3-7.

6- Prisma check list and other related files are fine and I have no word. The total writing style in this paper is good actually.

7- Publication bias and its likely impact should be well-indicated in the revised version.

8- While i am reading the discussion, I can not find the exact novelty of this research although I know how precious IS THIS research. Please indicate all novelty points of this research in new version.

9- Some of the references are outdatred, please revise them all.

10- Hpylori should be italic throughout the paper.

11- A minor polish for English is also needed.

Reviewer #2: Dear authors,

This is a meta-analysis dealing with the prevalence of CagA toxin and risk of Helicobacter pylori-infected gastric phenotype in Indo-Pacific region. The authors came to the conclusion that the distribution of CagA in Helicobacter pylori-infected gastric disorders varies throughout the eight Indo-Pacific countries, and CagA has a role in the development of gastric cancer.

The article is well-written however it does not add any new findings to the well known evidence regarding the relationship of Helicobacter pylori infection and gastric cancer.

Reviewer #3: I congratulate the authors for a good work, though it requires some modifications which I have included in the manuscript. Authors should also revise the introduction and make it a background and not sort of discussion.

6. PLOS authors have the option to publish the peer review history of their article (what does this mean?). If published, this will include your full peer review and any attached files.

Reviewer #1: No

Reviewer #2: No

Reviewer #3: **Yes: **JAMES Joseph YAHAYA

---

## [Author Response · Author response to Decision Letter 0]

16 Jun 2024

Reviewer #1: Dear Editorial office

It is my honour to read and review this nicely entitled paper by salomon et al “CagA toxin and risk of Helicobacter pylori-infected gastric phenotype: A meta-analysis of observational studies” submitted to Plos one. In principle I found it suitable for publication here pending your corrections.

Q1- The title seems fine and no change warranted.

Authors response

Thank you.

Q2- Abstract is too general-like, so please enrich it

Authors response

We have improved this section.

In the revised version

Background

 Helicobacter pylori (H.pylori) is frequently associated with non-cardia type gastric cancer, and it is designated as a group I carcinogen. This study aimed to systematically review and meta-analyse the evidence on the prevalence of CagA status in people with gastric disorders in the Indo-Pacific region, and to examine the association of CagA positive in the risk of gastric disorders. This study focused on the Indo-Pacific region owing to the high disability adjusted life-years related to these disorders, the accessibility of efficient treatments for this common bacterial infection, and the varying standard of care for these disorders, particularly among the elderly population in the region.

Methods

Relevant studies were identified in the health-related electronic databases including PubMed, Ovid, Medline, Ovid Embase, Index Medicus, and Google Scholar that were published in English between January 1, 2000, and November 18, 2023. For pooled prevalence, meta-analysis of proportional studies was done, after Freeman-Tukey double arcsine transformation of data. A random-effect model was used to compute the pooled odds ratio (OR) and 95% confidence interval (CI) to investigate the relationship between CagA positivity and gastric disorders.

Results

Twenty-four studies from eight Indo-Pacific countries (Bhutan, India, Indonesia, Malaysia, Myanmar, Singapore, Thailand, Vietnam) were included. Overall pooled prevalence of CagA positivity in H. pylori-infected gastric disorders was 83% (73-91%). Following stratification, the pooled prevalence of CagA positivity was 78% (95%CI:67-90%) in H. pylori-infected gastritis, 86% (95%CI = 73-96%) in peptic ulcer disease, and 83% (95%CI = 51-100%) in gastric cancer. Geographic locations encountered variations in CagA prevalence. There was a greater risk of developing gastric cancer in those with CagA positivity compared with gastritis (OR = 2.53,95%CI = 1.15-5.55).

Conclusion

Findings suggest that the distribution of CagA in H. pylori-infected gastric disorders varies among different type of gastric disorders in the study countries, and CagA may play a role in the development of gastric cancer. It is important to provide the high standard of care for the management of gastric diseases, particularly in a region where the prevalence of these disorders is high. Better strategies for effective treatment for high-risk groups are required for health programs to revisit this often-neglected infectious disease.

Q3- Abstract “To confirm these findings, well-designed studies with sufficient samples are recommended.” Is not absolutely a good text for here, remove it.

Authors response

We have removed it from the revised version. Thank you. 

Q4- Page 4, line 76, why there are two major types of reference methods? “epithelial cells [4],4 which” revise it for revision.

Authors response

It was a typing error. We have corrected it. Thank you.

Q5- Introduction section sentence “This reflects that, in

addition to its urease activity, H. pylori possesses additional virulence factors that are required to enable it to colonize and survive in the stomach's acid environment.” We need the below text references

** Abadi, A. T. B. (2017). Strategies used by helicobacter pylori to establish persistent infection. World journal of Gastroenterology, 23(16), 2870.

*** Israel, Dawn A., and Richard M. Peek Jr. "The role of persistence in Helicobacter pylori pathogenesis." Current opinion in gastroenterology 22.1 (2006): 3-7.

Authors response

 Thank you. We have added these two references (Ref# 4, 5). 

In the revised version

Q6- Prisma check list and other related files are fine and I have no word. The total writing style in this paper is good actually.

Authors response

Thank you.

Q7- Publication bias and its likely impact should be well-indicated in the revised version.

Authors response

We have provided an issue regarding publication bias along with a reference.

In the revised version

Study limitation

Due to publication bias related to small studies that tended to exaggerate the true effect [65], as well as limited number of studies that could compare CagA between the risks of gastric cancer and peptic ulcer disease, a type II error may be an issue. 

Reference

 65. Newcombe RG. Towards a reduction in publication bias. Br Med J (Clin Res Ed). 1987;295(6599):656-659

Q8- While i am reading the discussion, I cannot find the exact novelty of this research although I know how precious IS THIS research. Please indicate all novelty points of this research in new version.

Authors response

Agreed. There is not novelty in methodology. However, a specific evidence based on data from specific endemic countries in a particular region is informative. We have intended to provide a specific information of the targeted region. Hence, we have added these concerns in “implications”. 

In the revised version

Implications

Reiterating the substantial evidence of association between a treatable infection and a high burden of disease and health care utilisation can help improve equity in their programs during this second half of the Sustainable Development Goal period, when countries in the region are struggling to attain the universal health coverage goal set [20].

Q9- Some of the references are outdated, please revise them all.

Authors response

Thank you. We have used cotemporary references as well as a few old references to honour the original authors. References published in 2023 are also included. 

Q10- Hpylori should be italic throughout the paper.

Authors response

We have corrected it. Thank you.

Q11- A minor polish for English is also needed.

Authors response

We have improved language proficiency in this revised version. Thank you. 

Reviewer #2: Dear authors,

This is a meta-analysis dealing with the prevalence of CagA toxin and risk of Helicobacter pylori-infected gastric phenotype in Indo-Pacific region. The authors came to the conclusion that the distribution of CagA in Helicobacter pylori-infected gastric disorders varies throughout the eight Indo-Pacific countries, and CagA has a role in the development of gastric cancer.

The article is well-written however it does not add any new findings to the well known evidence regarding the relationship of Helicobacter pylori infection and gastric cancer.

Authors response

Thank you. We have intended to provide a specific information of the targeted region. We have noted in the revision that this study focused on people in the Indo-Pacific countries given the high DALY burden attributed to these disorders, the costs of management to the health system, the availability of effective treatments of this common bacterial infection and the variable quality of care for these disorders especially amongst older people in the region. We have added these points under “implication”.

Reviewer #3: I congratulate the authors for a good work, though it requires some modifications which I have included in the manuscript. Authors should also revise the introduction and make it a background and not sort of discussion.

Authors response

We have done corrections, following the points shown in the Manuscript. 

We have improved the introduction, as suggested. Thank you. 

Comments shown in the manuscript file of PONE-D-24-13758

Thank you for the comments and inputs provided. Below are the authors responses and subsequent texts in the revised version.

Q/1

The aims of the study don’t line up with the title of the study

Authors response

 We have rephrased the objectives to be clearer. We feel that the title is fit to the revised objectives.

In the revised version

Abstract

Background

This study aimed to systematically review and meta-analyse the evidence on the prevalence of CagA status in people with gastric disorders in the Indo-Pacific region, and to examine the association of CagA positive in the risk of gastric disorders.

Q2/ Specify them

 It should also be specified the time interval for the studies included in the study

Authors response

We have provided, accordingly.

In the revised version

Search strategy

Relevant studies were identified in the health-related electronic databases including PubMed, Ovid, Medline, Ovid Embase, Index Medicus, and Google Scholar that were published in English between 1 January 2000, and 18 November 2023.

Q3/

I suggest authors should use the word “pooled” instead of “summary

Authors response

We have updated, as suggested.

In the revised version

For pooled prevalence, meta-analysis of proportional studies was done, after Freeman-Tukey double arcsine transformation of data.

Q 4/Not disorders?

Authors response

We have corrected it, accordingly.

In the revised version

A random-effect model was used to compute the summary odds ratio (OR) and 95%

confidence interval (CI) to investigate the relationship between CagA and gastric 

disorders.

Q5/

This should be termed as “overall pooled prevalence

Authors response

We have updated, accordingly.

In the revised version

Overall pooled prevalence of CagA positivity in H. pylori-infected gastric disorders was 83% (95% CI=73-91%).

Q6/ This sentence must be rephrased. Use = symbol and not : symbol

 Compared with gastritis, there was a greater risk of developing gastric cancer in

those with CagA positivity (OR:2.53,95%CI:1.15-5.55)

Authors response

We have updated this part.

In the revised version

There was a greater risk of developing gastric cancer in those with CagA positivity compared with gastritis (OR = 2.53,95%CI =1.15-5.55).

Q7/ Conclusion

But this is not what is shown in the result section of the abstract. The variation of CagA positivity is not for countries but for different gastric disorders.

This information can’t be taken as a direct association. Authors should show plausibility and not causal-effect relationship

Authors response

Accordingly, we have updated this part. We have deleted the last sentence, following a comment by the Reviewer # 1.

In the revised version

Findings suggest that the distribution of CagA in H. pylori-infected gastric disorders varies among different types of gastric disorders in the study countries, and CagA may play a role in the development of gastric cancer.

Q 8/ These keywords cant help in searching of the work

Authors response

We have updated, accordingly.

In the revised version

Keywords:

Helicobacter pylori, CagA, gastritis, gastric cancer, peptic ulcers, meta-analysis, Indo-Pacific 

Q9/

Introduction....... This abbreviation must first be named before being used.

Authors response

We have provided, accordingly.

In the revised version

Helicobacter pylori (H.pylori) is a gram-negative bacterium that colonizes the stomach and is attributed to causing infections in humans [1.2].

Q 10/ Introduction

This information then contradicts with the statement given in the conclusion part of the abstract. After all, the number of gastritis or PUD is higher than that of gastric cancer. Although both have high prevalence of H. pylori.

Authors response

We have updated the abstract. Taking together with the reviewer #1’s comment, we have deleted the contradictory sentence in the abstract.

Q11/

Taken together………….what do you mean?

Authors response

We wish to say that taking all factors into consideration. We have rephrased this phrase in the revised version. 

Authors response

We have rephrased the objectives. 

In the revised version

Taking all of these into consideration, the objectives of this study were to systematically review and meta-analyse the evidence on the prevalence of CagA status in people with gastric disorders in the Indo-Pacific region, and to examine the association of CagA positive in the risk of gastric disorders.

Q12/ Materials and Methods

 what? This study sticked to the PRISMA-2020 checklist [22] [S1 Checklist]. 

The protocol of this……………Move this information to the declarations.

Authors response

We have replaced with a term “followed”. Thank you. 

In the revised version 

Materials and Methods

• This study followed the PRISMA-2020 checklist [22] [S1 Checklist].

Q13/ Search strategy

Correct the use of punctuation marks

Authors response

We have done this, accordingly. Thank you. 

Q14/Selection criteria

Did authors take into consideration issue of homogeneity for the included studies based on similarity of the detection methods of CagA?

 This is because all detection methods don’t have the same detection capacity

Authors response

We have considered studies regardless of the CagA detection methods. 

In the revised version 

Materials and Methods

Selection criteria 

3. Assessed the CagA status, irrespective of the detection method. 

Q15/

What is the message? Is there another way of considering this?

Authors response

We wish to say that we have followed the type of disorders according to the primary study. 

In the revised version 

Materials and Methods

Selection criteria 

In this study, the participants with duodenal and/or gastric ulcerations were categorised as peptic ulcer disease as stated in the primary studies.

Q16/ Data collection

Check grammar. How? This is not clear

Authors response

We have rephrased this part. 

In the revised version 

Full-text studies that deemed appropriate were further reviewed. Any discrepancy between the two investigators was settled by agreement.

Q17/ Table 2 Place references in column 2

Authors response

We have updated the table columns. 

In the revised version

 Table 1, 2

Q18/ Include reference

Authors response

We have provided a reference. 

In the revised version

 Stomach mucosa in the antrum and corpus almost always becomes infiltrated by neutrophilic and mononuclear cells during H. pylori colonization [56].

References

56. Kusters JG, van Vliet AH, Kuipers EJ. Pathogenesis of Helicobacter pylori infection. Clin Microbiol Rev.2006; 19:449-490.

---

## [Decision Letter · Decision Letter 1]

2 Jul 2024

CagA toxin and risk of Helicobacter pylori-infected gastric phenotype: A meta-analysis of observational studies

PONE-D-24-13758R1

Dear Dr. Naing,

We’re pleased to inform you that your manuscript has been judged scientifically suitable for publication and will be formally accepted for publication once it meets all outstanding technical requirements.

Kind regards,

Tomasz M. Karpiński, Prof.

Academic Editor

PLOS ONE

Additional Editor Comments (optional):

Reviewers' comments:

Reviewer's Responses to Questions

**Comments to the Author**

1. If the authors have adequately addressed your comments raised in a previous round of review and you feel that this manuscript is now acceptable for publication, you may indicate that here to bypass the “Comments to the Author” section, enter your conflict of interest statement in the “Confidential to Editor” section, and submit your "Accept" recommendation.

Reviewer #1: All comments have been addressed

Reviewer #2: All comments have been addressed

2. Is the manuscript technically sound, and do the data support the conclusions?

Reviewer #1: Yes

Reviewer #2: Yes

3. Has the statistical analysis been performed appropriately and rigorously? 

Reviewer #1: Yes

Reviewer #2: Yes

4. Have the authors made all data underlying the findings in their manuscript fully available?

Reviewer #1: Yes

Reviewer #2: Yes

5. Is the manuscript presented in an intelligible fashion and written in standard English?

Reviewer #1: Yes

Reviewer #2: Yes

6. Review Comments to the Author

Reviewer #1: well done

all revisions can help the authors to present the better version of the paper

now I have no extra comments for the authors

Reviewer #2: Dear authors, thank you so much for your efforts. The quality of the manuscript had improved muchly in the current version.

7. PLOS authors have the option to publish the peer review history of their article (what does this mean?). If published, this will include your full peer review and any attached files.

Reviewer #1: No

Reviewer #2: No

---

## [Editor Report · Acceptance letter]

9 Jul 2024

PONE-D-24-13758R1 

PLOS ONE

Dear Dr. Naing, 

I'm pleased to inform you that your manuscript has been deemed suitable for publication in PLOS ONE. Congratulations! Your manuscript is now being handed over to our production team.

Kind regards, 

on behalf of

Dr. Tomasz M. Karpiński 

Academic Editor

PLOS ONE